# Structural basis of phosphorylation-independent nuclear import of CIRBP by TNPO3

Qishun Zhou [1,7], Theo Sagmeister [2], Saskia Hutten[3], Benjamin Bourgeois[1], Tea Pavkov-Keller [2,4,5], Dorothee Dormann [3,6] & Tobias Madl [1,5] ✉

Transportin 3 (TNPO3) is a nuclear import receptor known for its broad substrate specificity, often recognizing arginine-serine (SR/RS) repeat-rich nuclear localization signals (NLS) in SRSF proteins. While serine phosphorylation or glutamate presence has been associated with these NLSs, recent proteomic studies identified TNPO3 cargoes lacking SR/RS repeats. One such example is the cold-inducible RNA-binding protein (CIRBP), which contains a non-classical RSY-NLS. Using X-ray crystallography, here we investigate the TNPO3-CIRBP interaction and find that tyrosines within the RSY-NLS play a key role in binding, independent of phosphorylation. Surprisingly, serine and tyrosine phosphorylation in CIRBP's NLS inhibits TNPO3 binding, suggesting a regulatory mechanism for nuclear import. Our study reveals a non-conventional nuclear import mechanism mediated by TNPO3, which may extend to other known or yet undiscovered TNPO3 cargoes.

Nuclear import is a crucial cellular process during which proteins are recognized by nuclear import receptors (NIRs), such as importins and transportins of the karyopherin β family[1–3]. These receptors function as active shuttles, transporting cargo between the cytosol and the nucleus through the nuclear pore complex[4]. Among the NIRs, transportin-3 (TNPO3) plays a vital role. Its deficiency leads to embryonal death in mice[5]. TNPO3 facilitates the nuclear import of various cargos, which regulate gene transcription, splicing, and polyadenylation, and even virus infections[6,7]. Typical cargos transported by TNPO3 include serine and arginine-rich splicing factors (SRSFs). These SRSFs are key regulators of gene transcription and splicing activities[6,8,9], reside in nuclear speckles[10], and impact cell proliferation[6], immune response[11], fat storage[12], and myogenesis[13]. Additionally, TNPO3 transports other cargos such as early B cell factor 1 (EBF1), a transcription factor involved in B-cell differentiation[14], and cleavage and polyadenylation specific factor 6 (CPSF6), a regulator of alternative polyadenylation[7]. TNPO3 also contributes to the nuclear import of Influenza A viruses (IAV) by assisting in their uncoating[15] and

regulates HIV replication by mediating the nuclear import of CPSF6[16,17]. Dysregulation of TNPO3-dependent nuclear import has substantial consequences. Mutations in TNPO3 cause limb-girdle muscular dystrophy D2 in humans[18–20], and conditional knockout of TNPO3 impairs T cell development in mice[5].

TNPO3 consists of 20 pairs of tandemly repeated alpha helices known as HEAT repeats, named after the proteins in which they were initially discovered[21]. It adopts a C-shaped structure enabling it to interact with the nuclear pore complex (NPC) on its outer surface and with cargos as well as RanGTP on its inner surface[1,22]. Upon binding to TNPO3, protein cargos are transported from the cytoplasm to the nucleus, a process that relies on energy-dependent recycling of NIRs[23,24]. This process, like in the case of other importins, subsequently involves migration of the TNPO3-cargo complex to the cell nucleus, release of cargo upon binding to RanGTP through direct competition, export of TNPO3 to the cytoplasm following a RanGDP/RanGTP gradient and the release of TNPO3 by hydrolysis of RanGTP to RanGDP[25–27]. Notably, TNPO3 is known to recognize nuclear

[1]Research Unit Integrative Structural Biology, Medicinal Chemistry, Otto Loewi Research Center, Medical University of Graz, Graz, Austria. [2]Institute of Molecular Biosciences, University of Graz, Graz, Austria. [3]Johannes Gutenberg Universität Mainz, Institute of Molecular Physiology, Mainz, Germany. [4]Field of Excellence BioHealth, University of Graz, Graz, Austria. [5]BioTechMed-Graz, Graz, Austria. [6]Institute of Molecular Biology (IMB) Mainz, Mainz, Germany. [7]Present address: Institut Pasteur, Université Paris Cité, CNRS UMR3528, Bacterial Transmembrane Systems Unit, Paris, France. ✉e-mail: tobias.madl@medunigraz.at

localization signals (NLSs) rich in serine-arginine (SR/RS) repeats[7,27]. Phosphorylation of serines within these SR/RS regions is thought to be crucial for efficient nuclear import, and is mediated by SR-specific protein kinases (SRPKs) and Cdc2-like kinases (CLKs)[28]. Structural studies of TNPO3 in complex with transcription factor SRSF1 (PDB ID: 4C0O) have revealed the recognition of an R-pS-R-pS-R motif by the HEAT repeats 14 to 17[27]. While proteomic studies have identified numerous potential TNPO3 cargos[27,29], most of them surprisingly lack consecutive SR/RS motifs typical for classic SR/RS-NLSs, suggesting the existence of alternative yet unknown NLSs recognized by TNPO3 (Fig. 1a, Supplementary Data 1, and ref. 29).

Recently, our colleagues, along with many contributors of this study, have identified a region rich in arginine, serine, and tyrosine (RSY) within CIRBP as an alternative NLS for TNPO3, that does not require phosphorylation[30]. Notably, while this RSY-NLS is also recognized by transportin-1 (TNPO1), its binding affinity is approximately 70 times lower than that of TNPO3. CIRBP, a nuclear protein known for its role in regulating mRNA stability and DNA repair, translocates from the nucleus to the cytoplasm under stress condition, forming stress granules[31,32]. It is a typical RNA-binding protein (RBP) harboring a folded RNA recognition motif and an RNA-binding RG/RGG region[33]. CIRBP's function is implicated in promoting cell proliferation and cancer development[34], highlighting its significance in understanding active nuclear import mechanisms, influencing its cellular functions.

Here in this study, our interest was to structurally characterize the RSY-NLS region of CIRBP bound to TNPO3, aiming to identify the key molecular determinants governing this interaction. We present the crystal structure (PDB: 8CMK) of the RSY region of CIRBP bound to TNPO3 at a resolution of 2.94 Å, revealing a previously unreported binding mode distinct from the canonical SR/RS-NLS. Our analysis underscores the essential role of three tyrosines within the RSY-NLS, engaging in multiple interactions with TNPO3, thus allowing efficient TNPO3 binding in vitro and nuclear import in vivo without requirement of prior phosphorylation. Through isothermal titration calorimetry (ITC) and cell-based hormone-induced nuclear import assays, we demonstrate that mutations affecting a single tyrosine residue within CIRBP's NLS abolish both TNPO3 binding and nuclear import. Surprisingly, and in contrast to classical NLSs, phosphorylation of the RSY region of CIRBP impairs TNPO3 binding, revealing a dual role of phosphorylation for different TNPO3 NLSs with opposing outcomes. Taken together, our findings offer mechanistic insights into alternative TNPO3 NLSs, potentially expanding our understanding of TNPO3-mediated nuclear import across various yet unexplored TNPO3 cargos.

## Results

### Phosphorylation of the RSY-NLS is dispensable for efficient TNPO3 binding

Phosphorylation of serine or glutamate substitution (mimicking phosphoserine) is required for TNPO3 recognition and associated nuclear import in canonical SR/RS-NLSs of SRSF1 and CPSF6, respectively[7,27,35]. A study has identified a $K_d$ of $1.3 \pm 0.19\,\mu M$ for phosphorylated SR/RS-NLS of SRSF1 and of $3.4 \pm 0.50\,\mu M$ for one of the phosphorylated putative SR/RS-NLS of CPSF6[7]. Strikingly, and in contrast to SRSF1, the RSY region of CIRBP (CIRBP$^{RSY}$) binds TNPO3 with nanomolar affinity in a phospho-independent manner, whereas no/weaker binding was reported for (un)phosphorylated SRSF1 and CPSF6, respectively[7,30]. Here we decided to use solution nuclear magnetic resonance (NMR) spectroscopy to perform the binding studies, aiming to reveal part of the structural details of TNPO3-CIRBP$^{RSY}$ interaction. We first carried out an examination of the TNPO3 binding affinities for the canonical (un)phosphorylated SR/RS-NLS from SRSF1 (SRSF1$^{SR}$) and unphosphorylated CIRBP$^{RSY}$. To obtain phosphorylated SRSF1$^{SR}$ we in vitro phosphorylated $^{15}N$ labeled SRSF1$^{RS}$ using recombinant SRPK1 and validated the phosphorylation efficiency of known SRSF1$^{SR}$ phosphosites using NMR spectroscopy. Please note, that both

CLK1 and SRPK1 have been reported to phosphorylate SRSF1[27,28]. Here, and as shown by NMR spectroscopy, SRPK1-mediated phosphorylation of SRSF1$^{SR}$ resulted in the appearance of several $^{1}H$-$^{15}N$ cross-peaks as a result of the chemical changes induced by the presence of the multiple phosphosites (pSRSF1$^{RS}$, Fig. 1b, c). This is consistent with previous observations showing that numerous serine residues within this fragment are phosphorylated by SRPK1[28,36]. We then titrated $^{15}N$-labeled SRSF1$^{SR}$ or pSRSF1$^{SR}$ with TNPO3. We observed only minor spectral changes of SRSF1$^{SR}$ cross peaks, indicating weak binding (Fig. 1b). In contrast, addition of TNPO3 to $^{15}N$-labeled pSRSF1$^{SR}$ caused extensive line broadening of several NMR signals thus confirming that phosphorylation of SRSF1 enhances TNPO3 binding (Fig. 1c). In line with the NMR data, the ITC-derived binding affinity of pSRSF1$^{RS}$ for TNPO3 was $2.42 \pm 0.33\,\mu M$, whereas no binding was detectable for the unphosphorylated version (Fig. 1d and Supplementary Table 1). This is in agreement with aforementioned previous FP studies[7]. In comparison, CIRBP$^{RSY}$ binds TNPO3 with an affinity of $0.61 \pm 0.10\,\mu M$ (Supplementary Fig. 1a and Supplementary Table 1).

Given that the sequence of the pSRSF1$^{SR}$ and CIRBP$^{RSY}$ TNPO3 binding motifs shows poor sequence similarity[27,30], we next aimed to answer the question if they interact with the same region of TNPO3 or not. We therefore compared the binding site of CIRBP$^{RSY}$, SRSF1$^{SR}$, and pSRSF1$^{SR}$ on TNPO3 using specific $^{13}C$-methyl labeled TNPO3. In this experiment, we made use of specifically labeled precursors of the amino acids isoleucine and methionine to label specific methyl groups which then act as NMR reporters of ligand binding. In the $^{1}H$-$^{13}C$ HMQC NMR spectra of [IM-methyl $^{13}C$] labeled TNPO3, two groups of cross-peaks show up corresponding to $^{13}C$ side chain methyl of methionine and isoleucine, respectively (Fig. 1e, lower left and upper right, respectively). As expected, no binding was observed between SRSF1$^{SR}$ and TNPO3, as indicated by the absence of chemical shift perturbations (CSPs) of TNPO3 methyl groups upon addition of unlabeled SRSF1$^{SR}$. However, and in line with the NMR data obtained for labeled cargo proteins, addition of either pSRSF1$^{SR}$ or CIRBP$^{RSY}$ to [IM-methyl $^{13}C$] labeled TNPO3 led to CSPs of several TNPO3 $^{1}H$,$^{13}C$ cross-peaks (Fig. 1e–g). This indicated direct interaction and confirmed that no phosphorylation is required for CIRBP$^{RSY}$ binding to TNPO3. Inspection of the NMR spectra revealed that both similar and distinct $^{1}H$-$^{13}C$ cross-peaks of isoleucine were affected by ligand binding, for which more peaks are affected at the addition of pSRSF1$^{SR}$. This indicates that pSRSF1$^{SR}$ and CIRBP$^{RSY}$ bind to partially overlapping binding sites on TNPO3 (Fig. 1f, g). Taking together, observations from three $^{1}H$-$^{13}C$ HSQC spectra directly suggested that CIRBP$^{RSY}$ binds to TNPO3 in a phospho-independent manner.

### CIRBP$^{RSY}$ binding to TNPO3 involves a non-canonical binding mode

We next aimed to gain insight into the structural basis for TNPO3 recognition of RSY-NLS and obtained crystals of TNPO3 C511A bound to CIRBP$^{RSY}$ that diffracted X-rays to 2.94 Å resolution. This TNPO3 mutant was used for the determination of the 3D X-ray structure of the TNPO3-SRSF1 complex and increases TNPO3 stability[27]. ITC control experiments showed a comparable binding affinity of CIRBP$^{RSY}$ for TNPO3 wild-type and C511A ($K_d$ of $0.61 \pm 0.10$ vs. $0.97 \pm 0.13\,\mu M$), respectively (Supplementary Fig. 1b). TNPO3 forms a dimer in the asymmetric unit of the crystal structure, as previously observed[37], and CIRBP$^{RSY}$ was bound to each monomer (Fig. 2a and Supplementary Fig. 2a). With an overall average B-factor of 86, the structure showed flexibility, still the vast majority of TNPO3 could be built into the electron density map with only some highly flexible loops missing. For the CIRBP$^{RSY}$ peptide, only 14 residues (amino acids 158–171) from the full CIRBP fragment (amino acids 138–172) could be modeled unambiguously. These residues form two short helices, each containing a single turn (residues 159–163 and 166–170; Fig. 2b). It is noteworthy, that a few contacts could be observed which are probably

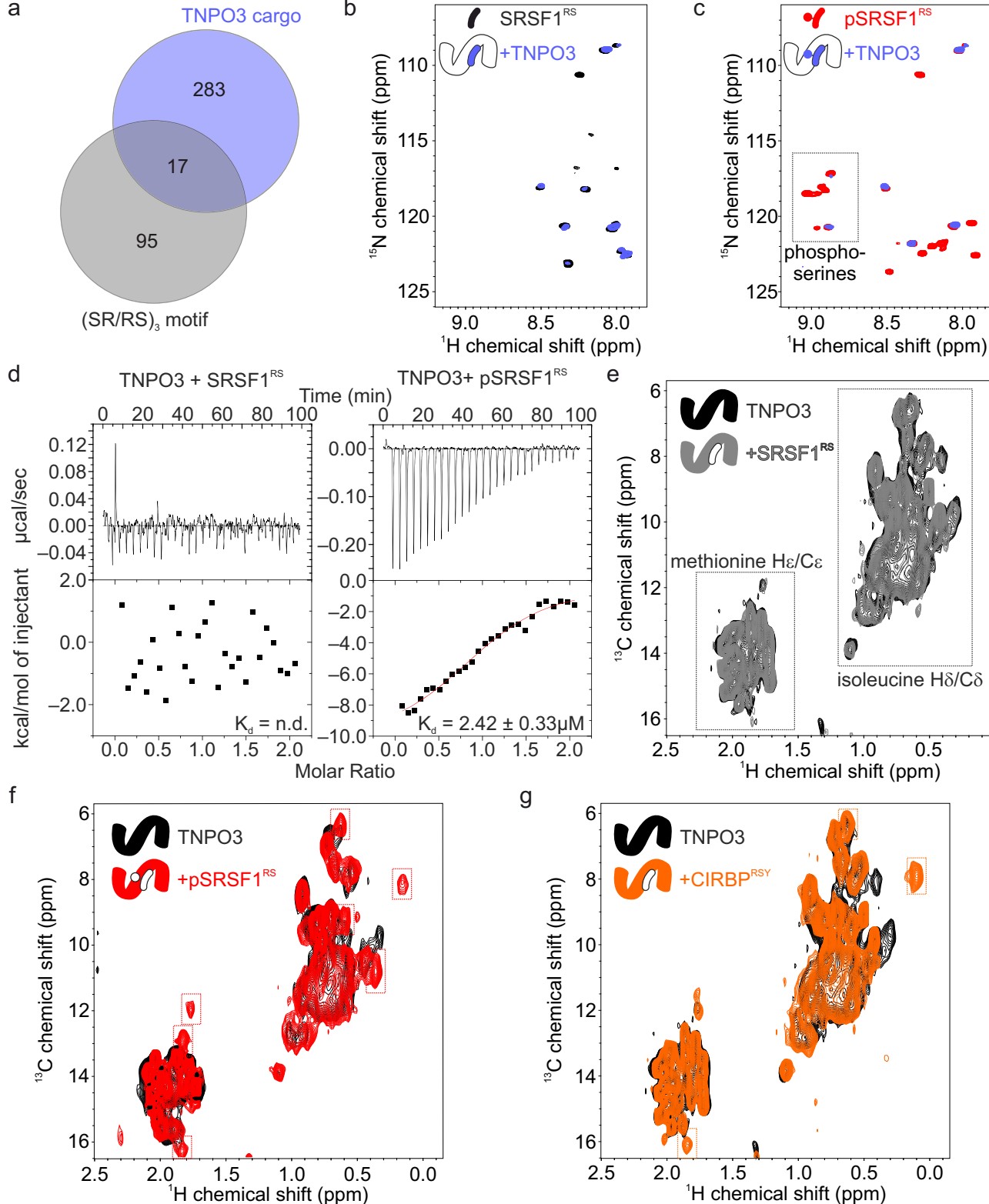

**Fig. 1 | Deciphering the modes of cargo binding to TNPO3. a** Venn diagram illustrating the overlap between putative TNPO3 binding partners identified in MS studies[29] and all human proteins containing the $(SR/RS)_3$ motif, encompassing all eight possible combinations of SRSRSR, SRSRRS, SRRSSR, etc. **b** Overlap of $^{1}H,^{15}N$ HSQC NMR spectra of 60 μM SRSF1$^{RS}$ alone (black) and with one stoichiometric equivalent of TNPO3 (blue). **c** Overlap of $^{1}H,^{15}N$ HSQC spectra of 60 μM phosphorylated pSRSF1$^{RS}$ alone (red) and with one stoichiometric equivalent of TNPO3 (blue). **d** ITC experiment titrating 100 μM SRSF1$^{RS}$ (left) or pSRSF1$^{RS}$ (right) into 10 μM TNPO3. The reported errors correspond to the SD of the fit. **e–g** $^{1}H,^{13}C$ HMQC spectra of 100 μM [IM-methyl $^{13}C$] labeled TNPO3 alone (black) and with one stoichiometric equivalent of SRSF1$^{RS}$ (**e**; gray), phosphorylated pSRSF1$^{RS}$ (**f**; red), or CIRBP$^{RSY}$ (**g**, orange), respectively. Peaks appearing upon ligand addition are indicated by dashed boxes.

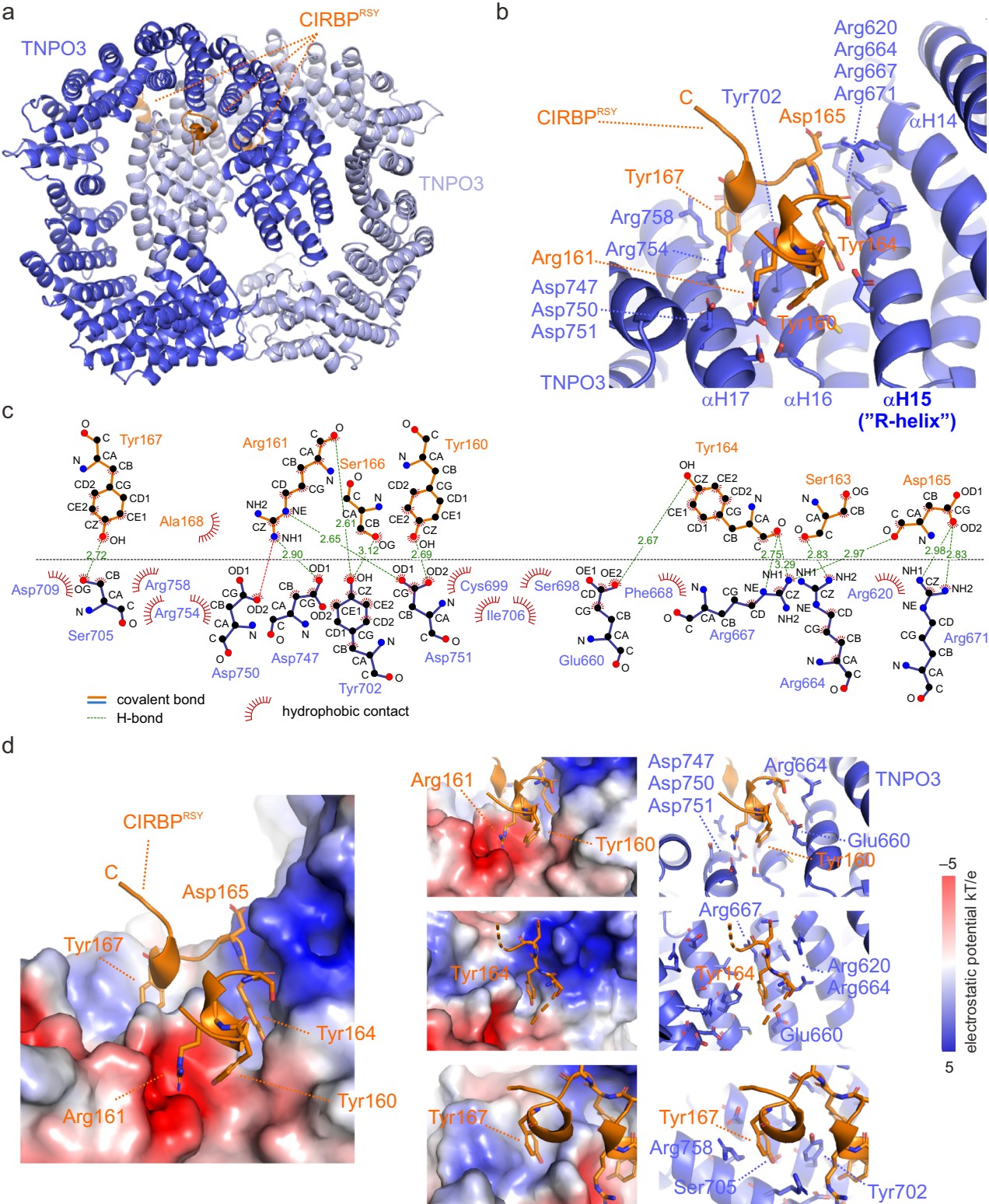

**Fig. 2 | Crystal structure of TNPO3-CIRBP^RSY complex. a** Overview of the crystal structure (PDB: 8CMK) showing the TNPO3 dimer in blue in complex with CIRBP^RSY in orange. **b** Zoom in to the binding site of TNPO3 and CIRBP^RSY, with the residues of the respective molecules colored in blue and orange. Residues that are involved in the interaction are indicated with the same color. The inner helix of TNPO3 HEAT repeat 15, which is refer to as R-rich helix is indicated. **c** Binding map showing the interactions details between individual residues of TNPO3 and CIRBP^RSY. **d** Electrostatic potential surface of TNPO3 at CIRBP^RSY binding surface with key residues highlighted. Figures of structure and electrostatic surface have been generated by Pymol, with PDB ID 8CMK, the contact map has been generated by LigPlot+[69,70].

crystallization artefacts, including (i) a third CIRBP fragment (amino acids 153–164) adjacent to TNPO3 chain B at the solvent-exposed outside of the protein (Supplementary Fig. 2b), (ii) contacts between the TNPO3 molecules in the dimer in proximity, but non-overlapping with the CIRBP[RSY] binding site (Supplementary Fig. 2c, d), and (iii) a single, though negligible, contact between the second TNPO3 molecule in the dimer and CIRBP[RSY] (Supplementary Fig. 2e). Among the intermolecular dimer contacts observed between TNPO3 molecules in chains A and B, the only reproducible interaction was a potential salt bridge between E304 and K877 (Supplementary Fig. 2c, d). Notably, TNPO3 dimerization was also observed in solution, with a dissociation constant of $8.3 \pm 1.0 \mu M$[37]. However, since endogenous TNPO3 concentrations are likely below this threshold, dimer formation is unlikely to be physiologically relevant under normal cellular conditions.

Inspection of the TNPO3-CIRBP[RSY] crystal structure revealed that the binding site of CIRBP[RSY] on TNPO3 is located between HEAT repeats 14 and 17 and involves more than 15 amino-acids (Fig. 2b–d and Supplementary Fig. 2a). Within CIRBP[RSY], several amino-acids between Y160 and Y167 mediate direct interactions with TNPO3 (Fig. 2b–d). The second TNPO3 molecule in the dimer showed a mostly identical binding mechanism, despite minor differences (Supplementary Fig. 2a). TNPO3-CIRBP complex formation is driven by several electrostatic interactions via the charged side chains of CIRBP R161 and TNPO3 D747/D751 as well as CIRBP D165 and TNPO3 R667/R671 (Fig. 2c, d). Importantly, three CIRBP tyrosines (Y167, Y160, and Y164) are engaged in a network of interactions including hydrogen bonds (with TNPO3 S705, D751 and E660), π stacking of CIRBP Y164 with TNPO3 Y702 and cation-π interactions of CIRBP Y167 and TNPO3 R758 (Fig. 2c, d and Supplementary Fig. 2f, g).

After having solved the crystal structure of the CIRBP[RSY]–TNPO3 complex, we compared the binding modes of canonical (pSRSF1, pCPSF6) that have been previously published[7,27], and the non-canonical ligand CIRBP to shed light on the distinct binding mechanism. Unlike the canonical ligands pSRSF1 and pCPSF6, which lacks a helix at the binding site, CIRBP[RSY] forms two short helices between S159-S163 and S166-H170, respectively (Fig. 3a, b). To accommodate the loop connecting these helices, the TNPO3 ligand-binding cavity opened up which is visible by a movement of TNPO3 helices 14 and 15. Detailed analyses of the three crystal structures revealed shared contacts between the different ligands and distinct contacts. Shared contacts involved the electrostatic interactions formed by TNPO3 R667 (CIRBP-D165, SRSF1-pS207, CPSF6-E523), R671 (CIRBP-D165, SRSF1-pS209, CPSF6-pS525), D750 (CIRBP-R161, SRSF1-R206, CPSF6-R522), and D751 (CIRBP-R161, SRSF1-R206/208, CPSF6-R522/524; Fig. 3c). Contacts distinct for CIRBP are (i) electrostatic interactions between CIRBP R161 and TNPO3 D747, (ii) hydrogen bonding between CIRBP S163 and TNPO3 R664, as well as CIRBP Y164 and TNPO3 E660, (iii) hydrogen bonding between CIRBP S166 and TNPO3 Y702, and (iv) cation-π interactions between CIRBP Y167 and TNPO3 R758 (Fig. 3d and Supplementary Fig. 2g).

In order to further evaluate these findings, we obtained synthetic peptides corresponding to the C-terminal end of the CIRBP RSY region (157-172, referred to as mCIRBP[RSY]) found in the crystal structure of the CIRBP-TNPO3 complex. Additionally, we generated 3 mutants, each with a single point mutation at either Y164A, Y167A or R161A. These residues displayed either hydrogen bonding and π stacking (Y164), hydrogen bonding (Y167), or a combination of interactions (R161). In agreement with our 3D structure of the complex, all mutations resulted in impaired binding to TNPO3 C511A, as observed using ITC (Table 1, Supplementary Fig. 3a, and Supplementary Table 1).

In summary, the 3D structure of the CIRBP[RSY]–TNPO3 complex showed that the interaction is mediated by both ionic interactions involving charged residues and hydrogen bonds involving the tyrosines of the CIRBP NLS. The latter binding mechanisms is specific to the RSY-NLS and not been found present in canonical TNPO3 NLSs.

Recognition of the CIRBP tyrosines compensates the lack of phosphorylation via structural arrangements of the TNPO3 binding surface and generation of binding pockets accommodating tyrosine residues.

## CIRBP-like motifs are frequent among IDRs, including TNPO3 cargos

We used the ScanProsite tool[38] to search for CIRBP-like NLSs in the human proteome to identify yet unknown TNPO3 cargos. The core TNPO3 binding motif of CIRBP Y-R-x-S-Y-D-S-Y is highly conserved in RBMX (RNA-binding motif protein, X chromosome), despite not being among the currently identified TNPO3 cargos (Fig. 4a, b). Because most of the contacts involve the tyrosines and arginines in the CIRBP motif, we then scanned a simplified motif Y-R-x(2,3)-Y-x(2,3)-Y, in which x(2,3) corresponds to either two or three linker residues. This analysis identified over 90 proteins, 33 of which contain the motif within their intrinsically disordered regions (IDRs) according to the AlphaFold2 structures available from UniProt (Supplementary Data 2)[39]. AlphaFold2 binding prediction of all these motifs to TNPO3 yielded that the great majority of complexes are forming through the CIRBP-binding site, including 23 having all the five models converged to the same site and six partly converged. Only four showed diverged binding sites (Supplementary Table 2). As a control, we ran AlphaFold2 predictions of the known TNPO3-NLSs CIRBP[RSY], SRSF1[RS,] and CPSF6[RS]. While these known NLSs yielded relatively low IDDT scores, Alpha-Fold2 correctly predicted their binding locations. Importantly, none of the 33 newly identified proteins have previously been reported as TNPO3 cargo[27,29].

From our X-ray structure of the CIRBP/TNPO3 complex, it seemed that the CIRBP tyrosines contribute hydrogen bonds, π-stacking and cation-π interactions to strengthen the interaction (Fig. 2c and Supplementary Fig. 2a, f, g). To investigate whether other aromatic amino acids could also allow binding, we scanned for a [YWF]-R-x(2,3)-[YWF]-x(2,3)-[YWF] motif that accepts phenylalanine or tryptophan as alternative of tyrosine. Our scan resulted in 1343 proteins, of which 297 are nuclear proteins (Fig. 4a and Supplementary Data 3 and 4). We compared the list of proteins harboring the motif with the list of previously identified TNPO3 cargos[29] and found that 12 overlapped (Fig. 4a) out of which 3 harbor the motif within IDRs, respectively delta-1-pyrroline-5-carboxylate synthase (P5CS), zinc finger CCCH domain-containing protein 18 (ZCH18), and RNA-binding protein 39 (RBM39; Fig. 4b). We tested these three putative TNPO3 binding motifs and the one of RBMX, which is highly conserved with CIRBP[29], for their TNPO3 binding. [1]H NMR spectra clearly indicated line broadening of the [1]H signals of all the tested peptides upon addition of TNPO3, thus indicating direct interaction (Fig. 4c). In P5CS, the disappearance of the tryptophan signal corresponding to aromatic [1]Hζ2 at 7.6 ppm suggests that an aromatic cycle other than tyrosine can also bind to TNPO3. In the predicted binding complex by AlphaFold2, all peptides except P5CS are binding to the same binding site as CIRBP[RSY] (Fig. 4c and Supplementary Table 2). Taking together, these regions rich in aromatic cycles are likely to interact with TNPO3 as potentially distinct NLSs.

## Mutation of the key RSY-NLS residues impairs CIRBP nuclear import

We demonstrated that the absence of one key arginine or tyrosine in CIRBP[RSY] was sufficient to disrupt TNPO3-binding in vitro. To further test if reduced binding of those mutants to TNPO3 also results in reduced nuclear import, we employed our well-established hormone-inducible nuclear import assay in HeLa cells[30,40]. By fusion of full-length CIRBP to two hormone-binding domains of the glucocorticoid receptor (GCR), CIRBP is retained in the cytoplasm upon transient transfection. Upon addition of the steroid hormone dexamethasone, it is released, and its nuclear import can be analyzed by quantitative live cell imaging due to the fusion of the GCR-reporter to GFP ($GCR_2$-$GFP_2$-

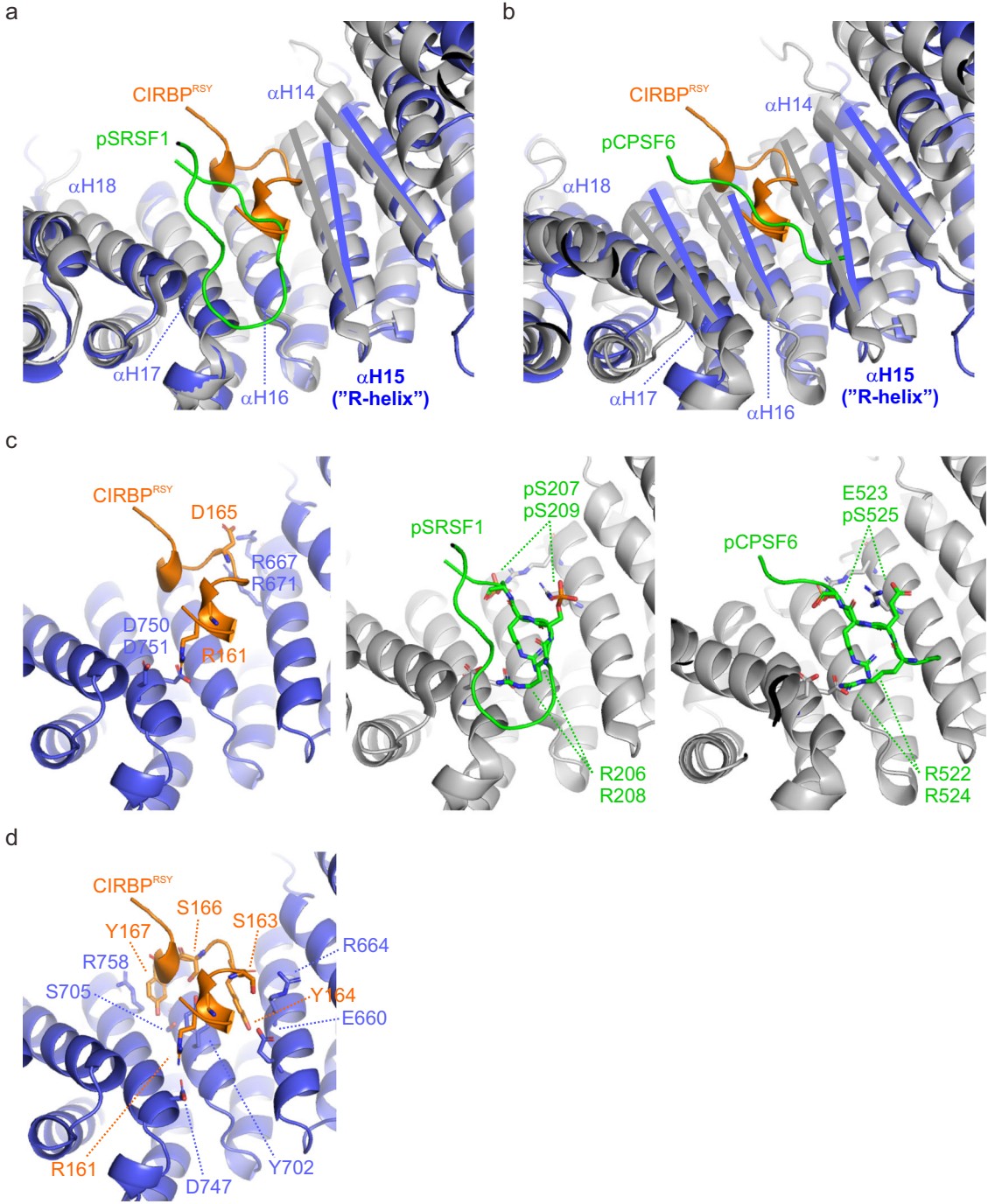

**Fig. 3 | Comparison of the binding modes in the TNPO3-CIRBP^RSY (PDB: 8CMK), TNPO3-pSRSF1^RS (PDB: 4C0O), and TNPO3-pCPSF6^RS (PDB: 6GX9) complexes. a, b** Overlap of the TNPO3-CIRBP^RSY (blue-orange) and TNPO3-pSRSF1^RS (gray-green, **a**) and TNPO3-CIRBP^RSY and TNPO3-pCPSF6^RS (gray-green, **b**) complexes, respectively. The inner helix of TNPO3 HEAT repeat 15, corresponding to R-rich helix is indicated. **c** Details of the binding interfaces of the three complexes highlighting charged residues that form shared electrostatic interactions in each of the complexes. **d** TNPO3-CIRBP^RSY binding interfaces highlighting the residues mediating contacts which are specific for CIRBP.

CIRBP). (Fig. 5a). As shown in Fig. 5b (and Fig. 5c for quantification), CIRBP wildtype (WT) was efficiently imported in the nucleus within 30 min after dexamethasone addition. Interestingly however, both the CIRBP R161A and Y164A mutants, were strongly impaired in nuclear import, comparable to the GCR-reporter lacking any NLS (GCR$_2$-GFP$_2$ empty). These findings indicate that the absence of an individual key residue in the TNPO3 binding motif of CIRBP^RSY indeed abrogates its nuclear import, and our ITC results are consistent with observations in the cellular context.

## Phosphorylation of CIRBP^RSY impairs TNPO3 binding
Serine phosphorylation by either SRPK1 or CLK1 is essential for the nuclear import of proteins with canonical SR/RS-NLSs[27]. Although the TNPO3 binding motif of CIRBP^RSY does not resemble conventional SR/RS repeats, residues Y164, S166, and Y167 have been identified to be phosphorylated in various biological contexts[41]. To examine the impact of phosphorylation on TNPO3 binding, we first performed an in vitro phosphorylation assay of $^{15}$N-labeled CIRBP^RSY using recombinantly purified SRPK1. Over time, we observed the appearance of three

**Table 1 | Summary of the ITC results of TNPO3 binding studies with different CIRBP$^{RSY}$ fragments and mutants**

| CIRBP motif | Sequence | Affinity |
|---|---|---|
| WT | GGSYRDSYDSYATHNE | $K_d = 5.78 \pm 1.63\ \mu M$ |
| R161A | GGSYADSYDSYATHNE | No binding |
| Y164A | GGSYRDSADSYATHNE | No binding |
| Y167A | GGSYRDSYDSAATHNE | Low affinity |

$^1$H,$^{15}$N cross peaks located in a region characteristic for phosphorylated serines (Fig. 5d)[42]. The kinetics of appearance of these NMR signals inversely correlated with the disappearance of three peaks previously assigned to the unmodified serines S146, S148, and S152 (BMRB ID 28025; Fig. 5d, e). Strikingly, and in contrast to canonical import, phosphorylation of these serines significantly decreases the binding affinity of in vitro phosphorylated CIRBP$^{RSY}$ to TNPO3 ten-fold, from a $K_d$ of $0.61 \pm 0.10$ to $5.99 \pm 0.96\ \mu M$ (Fig. 5f). This reduction was comparable to the $K_d$ of the core CIRBP$^{RSY}$ binding motif (residues 157–172) binding to TNPO3, i.e. $5.78 \pm 1.63\ \mu M$ (Supplementary Fig. 3a), indicating that although the N-terminal part of the CIRBP$^{RSY}$ was not resolved in the crystal structure, it may also contribute to TNPO3 binding. In line with these observations, $^1$H,$^{13}$C NMR cross peaks of [IM-methyl $^{13}$C] labeled TNPO3 shifted to comparable end points upon addition of CIRBP$^{RSY}$ or pCIRBP$^{RSY}$, respectively (Fig. 5g and Supplementary Fig. 4a). Taken together, it is unlikely that SRPK1 phosphorylates any of the residues within the core binding motif of CIRBP. Since the kinase(s) catalyzing formation of these phosphosites are unknown, we ordered synthetic peptides of CIRBP (157–172) containing a phosphorylated variant of each of the three residues. ITC analysis showed that all three phosphorylation events impaired binding to TNPO3 C511A. Peptides containing pY164 or pS166 decreased binding beyond being detectable or fittable (Supplementary Fig. 3b).

These results indicate that the nuclear import of CIRBP by TNPO3 via the tyrosine-rich motif is negatively regulated by serine and tyrosine phosphorylation. Despite the yet unidentified serine/tyrosine kinase involved, it is evident that phosphorylation of each of these residues results in a substantial reduction in nuclear localization. This stands in contrast to the classical mechanism of TNPO3-mediated nuclear import, which is promoted by phosphorylation[27], suggesting a divergent mechanism possibly involving a distinct class of cargo proteins.

## Discussion

We employed a combination of biophysical techniques and X-ray crystallography to investigate the intricate molecular interactions between TNPO3 and the non-classical NLS within CIRBP. Our investigation revealed that this binding hinges on a concise motif rich in tyrosines, where both arginine and multiple tyrosines play pivotal roles in facilitating these interactions. However, despite the conservation of the binding site in TNPO3, this mode of binding represents a non-conventional mechanism that is disrupted by the phosphorylation of serine and tyrosine residues within the NLS.

Previous studies examining the complexes of TNPO3 with CPSF6 and SRSF1, respectively, displayed an overlapped binding surface, with the core binding motifs exhibiting high comparability[7,27]. In contrast, when comparing the positions of the phospho-residues in the canonical complexes to those of CIRBP residues, we observed no direct structural overlap. For instance, CIRBP Y160 and A168 are located near SRSF1 pS207/pS209, while CIRBP R161 aligns closely with CPSF6 pS525. In the classical TNPO3-SRSF1$^{RS}$ complex, Y702 forms a hydrogen bond with phosphoserine pS209[27], whereas for the TNPO3-CIRBP$^{RSY}$ complex, Y702 engages in hydrogen bonds with R161 and S166 through its aromatic hydroxyl group. Additionally, it participates in hydrophobic interactions, including π stackings, via its aromatic ring, positioning Y702 in between Y160, R161, Y164, S166, and Y167 (Fig. 2b). The presence of Y702 and the inward fold of R667 causes a cavity on TNPO3 surface, proximate to Y702, enabling interaction with the three crucial tyrosines of CIRBP$^{RSY}$ (Fig. 2b, d). These cavities allow the hydroxyl group of tyrosine to form hydrogen bonds with neighboring residues such as serine, glutamate, or aspartate. Notably, a comparable cavity is absent in the complexes with SRSF1 or CPSF6, despite both displaying a tyrosine located at the periphery of the binding surface[7,27]. This suggests that this atypical binding mode allows for unphosphorylated, tyrosine-rich cargos in addition to conventional RS/SR regions.

The distinction between these two mechanisms is further underscored by the involvement of hydrogen bonds formed by -OH groups of tyrosine, serving as an additional facilitator of binding. Unlike forming salt bridges with multiple arginines in TNPO3, tyrosines can interact with the backbone of various amino acids, including arginine, as well as with side chains of polar amino acids such as serine, aspartate, and glutamate (Fig. 2c). This nuanced interaction pattern suggests that the presence of tyrosine does not merely substitute the serine phosphate group but rather leads to the formation of alternative binding modes. This observation could explain why phosphorylation is dispensable in the tyrosine-rich motif. Intriguingly, mutating a single tyrosine substantially impairs binding of CIRBP to TNPO3 in vitro and nuclear import of CIRBP in cells. Consequently, there is compelling evidence to assert that the binding mechanisms of TNPO3-SRSF1$^{RS}$ and TNPO3-CIRBP$^{RSY}$ diverge substantially (Figs. 2b and 3a–c). Given these findings, an independent future investigation into the consensus NLS sequence based on the latter structure is considerably necessary.

Based on our X-ray structure of the CIRBP$^{RSY}$-TNPO3 complex, we derived Y-R-x-S-Y-D-S-Y as a non-canonical TNPO3 binding motif (Fig. 2c). This specific motif is exclusive to CIRBP, although RBMX and RMXL1 (RNA binding motif protein, X-linked-like-1, see Supplementary Table 2) exhibit a high sequence similarity with only the aspartate being replaced by a glutamate. Despite not being previously identified as TNPO3 cargoes, we demonstrated through NMR spectroscopy and AlphaFold2 predictions that the RBMX/RMXL1 motifs directly interact with TNPO3 (Fig. 4c). In our search for other TNPO3 cargos resembling CIRBP-like NLS motifs, we conducted a scan of the human proteome. We first focused on identifying a simplified motif, Y-R-x(2,3)-Y-x(2,3)-Y, containing all essential arginine and tyrosine residues. This search yielded 33 proteins, including 16 nuclear proteins (such as RBMX and RMXL1) and 17 non-nuclear proteins, according to UniProt[39]. Non-nuclear proteins appearing in this list may arise as genuine false positives, proteins with TNPO3 functions unrelated to nuclear import, such as TNPO3's chaperoning role in cytosolic stress granules, or due to insufficient information available for these proteins[30,43,44]. Many of these protein motifs are enriched in glutamate, aspartate, serine or threonine in addition to arginine/tyrosine, which might further form additional contacts. It is important here to note that in AlphaFold2 predictions we only analyzed the models in terms of the binding sites and refrained from more detailed interpretation due to the low PLDDT scores observed for most of the peptide ligands. Despite these proteins not being previously identified as TNPO3 binding partners in proteomics screens[27,29], our structural predictions suggest them to be unrecognized TNPO3 cargos. We expanded our analysis to include the other aromatic residues phenylalanine and tryptophane in place of tyrosine, resulting in the [YWF]-R-x(2,3)-[YWF]-x(2,3)-[YWF] motif. This motif was found in 297 nuclear proteins (Fig. 4a), with four of them previously identified as TNPO3 cargos. Additionally, we found eight non-nuclear proteins bearing this motif to interact with TNPO3 (Supplementary Fig. 5 and ref. 29) and confirmed binding of predicted targets using NMR spectroscopy. In summary, our analyses identified a substantial number of putative TNPO3 cargos in the human proteome. However, the binding affinities of the respective motifs, as well as their biological significance, remain subjects of ongoing investigation.

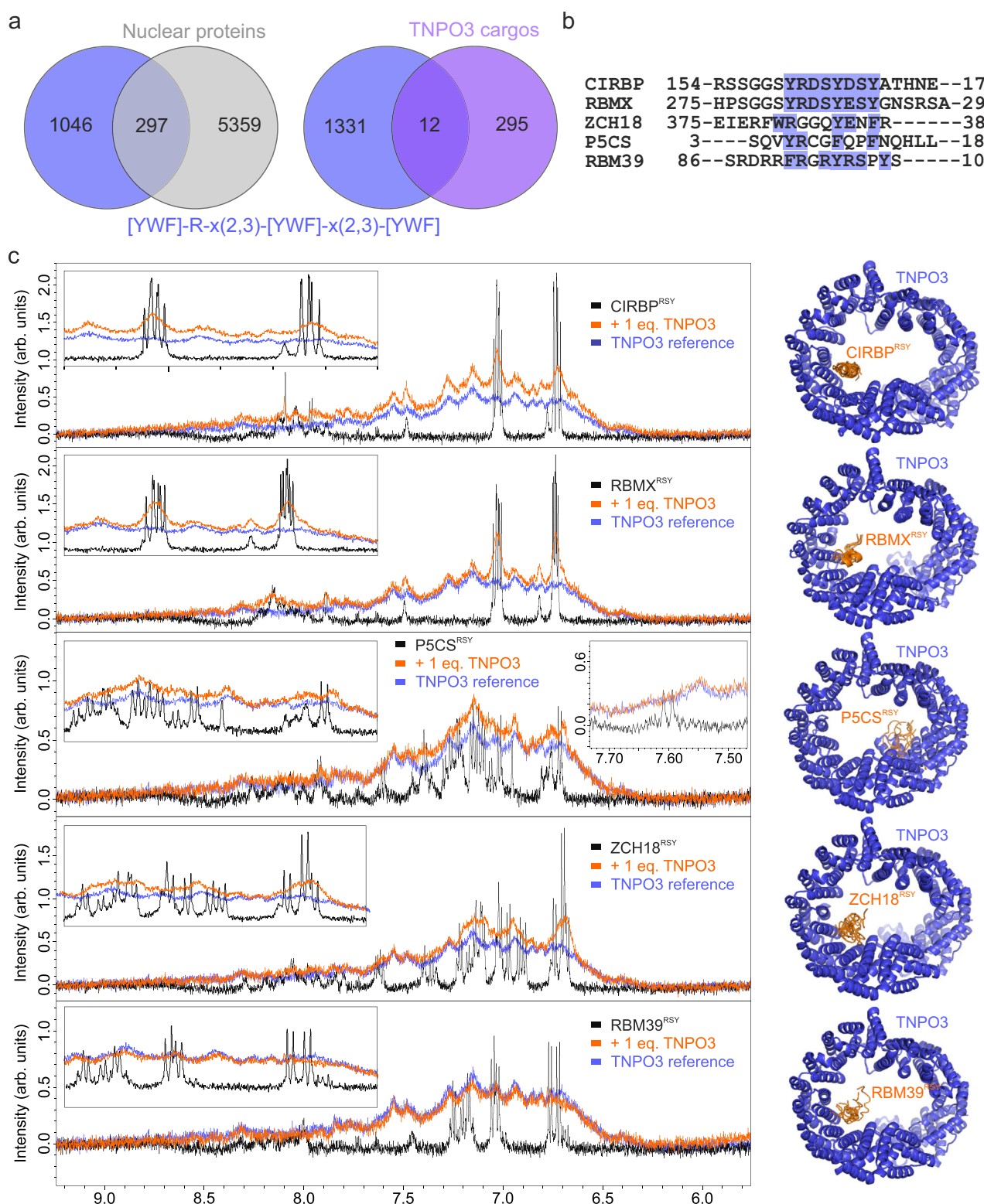

**Fig. 4 | Interaction studies of TNPO3 and its putative cargos. a** Venn diagram showing the overlap of proteins containing the search motif with human nuclear proteins or with proteins reported as TNPO3 cargos[29], respectively. **b** Overlap of putative IDRs harboring the [YWF]-R-x(2,3)-[YWF]-x(2,3)-[YWF] motif, including CIRBP and RBMX. **c** Overlay of ¹H NMR spectra of CIRBP (157–172), RBMX^RSY (276–291), P5CS^RSY (3–20), ZCH18^RSY (371–390), RBM39^RSY (86–101) free (black), in presence of a stoichiometric amount of TNPO3 (orange), with corresponding AlphaFold2 prediction of the complex structure. The inlays show a zoom of key spectral regions of the cargo peptides which become broadened upon addition of TNPO3. All experiments were carried out using either 50 μM of the peptide, TNPO3 or both, respectively. In all overlays, the spectrum of TNPO3 alone is shown in blue as a reference. For all AlphaFold2 predictions an overlay of 5 independent calculations is shown.

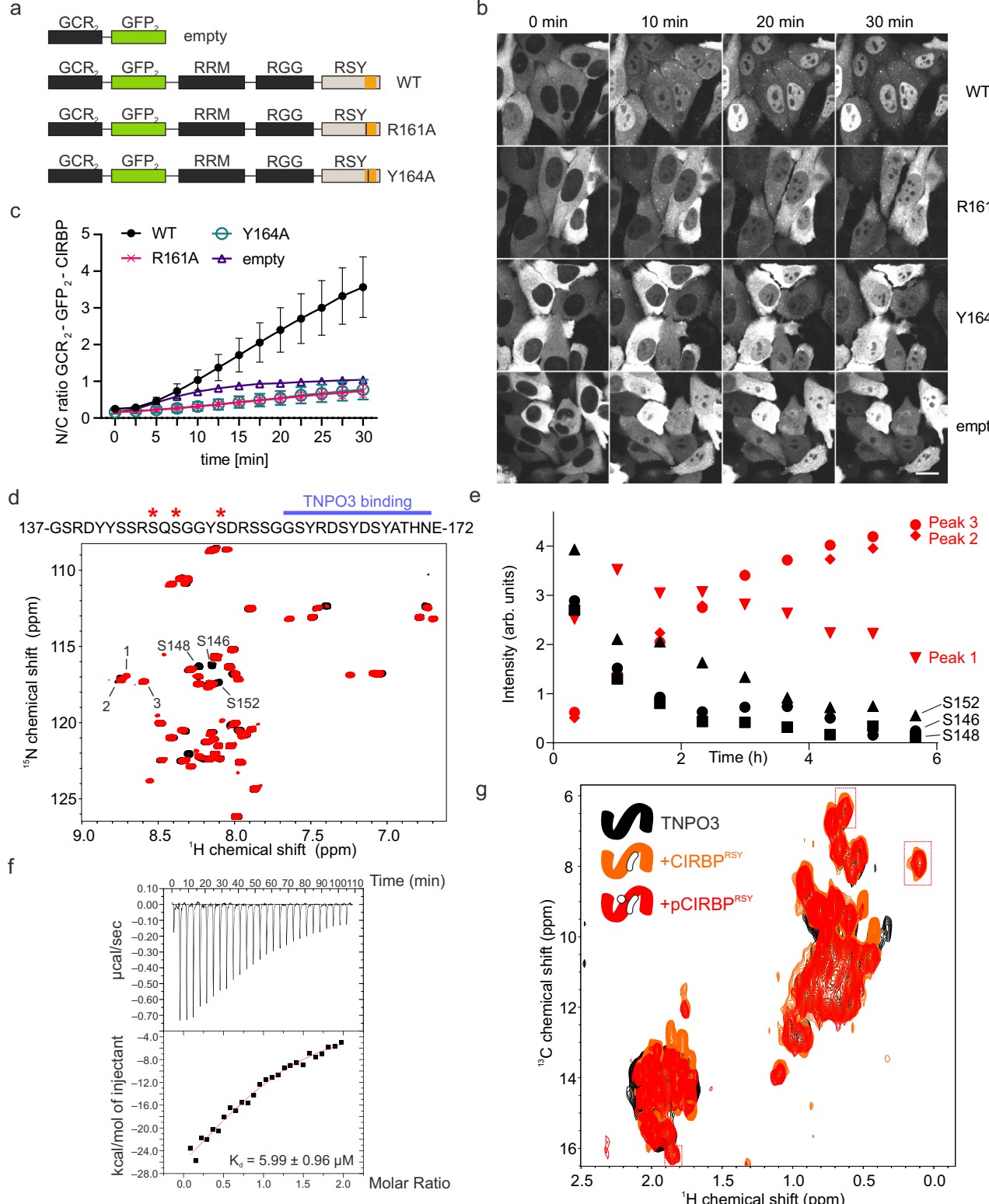

**Fig. 5 | Cell-based and phosphorylation studies of CIRBP. a** R161A and Y164A mutations disrupt CIRBP nuclear import in a hormone-inducible import assay. Constructs of different GCR$_2$-GFP$_2$-CIRBP reporters, with the TNPO3 binding motif in the CIRBP$^{RSY}$ domain highlighted in orange. **b** Representative images of timelapse imaging for GCR$_2$-GFP$_2$-CIRBP wildtype (WT), R161A or Y164A mutants or the empty reporter lacking the CIRBP sequence (empty) in Hela cells. Bar, 20 μm. **c** Quantification of nuclear import from (**b**) by calculating nuclear/cytoplasmic (N/C) fluorescence intensity ratio. Mean SD of three independent replicates with at

least 40 cells each (two replicates for the empty vector). **d** Overlay of $^1$H-$^{15}$N HSQC NMR spectra of $^{15}$N-labeled CIRBP$^{RSY}$ at the beginning and end of the in vitro phosphorylation by SRPK1. **e** Quantification of (**d**). **f** ITC curve showing a titration of 100 μM phosphorylated pCIRBP$^{RSY}$ into 10 μM of TNPO3. The reported errors correspond to the SD of the fit. **g** Overlap of $^1$H-$^{13}$C HMQC spectra corresponding to 100 μM TNPO3 alone (black), with 1 stoichiometric equivalent of CIRBP$^{RSY}$ (orange) or pCIRBP$^{RSY}$ (red), respectively. Peaks that appeared upon addition of the respective ligand are indicated. Source data are provided in Source data file.

Serine and tyrosine phosphorylation are pivotal in various cellular functions, including signaling, cell cycle regulation, immune responses, DNA damage responses, and cell proliferation[45–50]. Particularly within these processes, phosphorylation acts as a key regulator of protein nuclear translocation, hence regulating the function of the associated proteins[51–53], such as TNPO3 cargos like SRSF1 and CPSF6[7,27]. CIRBP is a protein whose subcellular localization is vital for its function[30,54]. Consequently, the regulation of its nuclear import by TNPO3 through serine and tyrosine phosphorylation is of great importance in determining its function. Unexpectedly, phosphorylation of CIRBP$^{RSY}$ caused an opposing response compared to phosphorylation of the classical NLS: serine phosphorylation of CIRBP$^{RSY}$ by SRPK1 (pS146, pS148, and pS152) decreased its binding to TNPO3 by 10 fold, while phosphorylation of tyrosine and serine residues at the C-terminus of CIRBP$^{RSY}$ (Y164, S166, and Y167) completely abolished TNPO3 binding (Fig. 5f and Supplementary Fig. 3b). These sites (Y164, S166, and Y167) are distinct from the RS/SR motif and are not phosphorylated by SRPK1[55,56]. Therefore, it is plausible that an unknown tyrosine kinase might negatively regulate the nuclear import of CIRBP by phosphorylating tyrosine residues at its C-terminus. This regulatory mechanism could be associated with a protein family binding to TNPO3 through a tyrosine-rich region, thereby inhibiting various cellular processes by preventing the responsible protein from localizing into the nucleus. Future investigations will focus on exploring the conservation of tyrosine-mediated nuclear import in the human proteome and identifying the kinase(s) responsible for phosphorylating the corresponding tyrosine(s).

## Methods

### Plasmids
All plasmids were purchased from Genscript. Expression constructs for wild-type (WT) human CIRBP$^{RSY}$ (138–172) and full-length (FL) WT human TNPO3 were generated by synthesizing codon-optimized DNA fragments and cloning them into a pETM11-His$_6$-protein A expression vector. This vector contains a Tobacco Etch Virus (TEV) protease cleavage site downstream of the protein A tag and was previously used in an earlier study[30]. Expression constructs for WT human SRSF1$^{RS}$ (195-211), full-length WT human SRPK1 and human TNPO3 C511A mutant were inserted into a pETZ2_1a-His$_6$-protein A vector.

Additional constructs ligated into the GCR$_2$-GFP$_2$ expression vector were also synthesized and purchased from GenScript.

### Protein expression and purification
For expression of recombinant unlabeled or isotopically labeled His$_6$-protein A tagged protein constructs, the different bacterial expression vectors were transformed into *Escherichia coli* (*E. coli*) BL21-(DE3) Star strain (Invitrogen, ref. C601003). During expression of TNPO3 FL WT and C511A mutant, 1 mL preculture was inoculated to 1 L minimal medium supplemented with 6 g of glucose (Carl Roth GMBH) and 1 g of $^{15}$NH$_4$Cl (Sigma) for $^{15}$N-labeled proteins or 3 g of NH$_4$Cl (Sigma) for unlabeled proteins. 1 L culture was grown at 30 °C with for 72 h followed by dilution with 1 L of the same medium. Cultures after dilution were incubated at 30 °C for an additional 1.5 h before adding 1 mL 0.5 M isopropyl β-D-1-thiogalactopyranoside (IPTG) to induce the expression. 50 mg methionine and α-ketobutyrate (precursor of isoleucine) with $^{13}$C-methyl labeling of the side chains were added to $^{15}$N-labeled TNPO3 before induction with IPTG. Protein expression was carried out at 20 °C for 16 h. For SRPK1 FL and disordered CIRBP$^{RSY}$/SRSF$^{RS}$, 10 mL preculture was inoculated in 1 L lysogeny broth (LB) medium for unlabeled proteins followed by growing at 37 °C until OD (600 nm) reached 0.8. Cultures were further induced with a final concentration of 0.5 mM IPTG followed by protein expression for 16 h at 20 °C. Cell pellets were harvested, resuspended and sonicated either in denaturing lysis buffer (50 mM Tris–HCl pH 7.5, 150 mM NaCl, 20 mM Imidazole, 6 M urea) for disordered protein fragments (CIRBP$^{RSY}$, SRSF1$^{RS}$) or in non-denaturing lysis buffer (50 mM Tris–HCl pH 7.5, 150 mM NaCl, 20 mM imidazole, 2 mM tris(2-carboxyethyl)phosphine [TCEP]) for folded proteins (SRPK1, TNPO3 and TNPO3 C511A). 100 µL bacterial protease inhibitor (Carl Roth, containing 100 mM AEBSF, 5 mM Bestatin, 1.5 mM E-64, 0.2 mM Pepstatin A, and 2 mM Phosphoamidon) has been additionally added into the resuspension of SRPK1 pellet from 2 L medium before sonication. After sonication at 70% amplitude and 1 s pulse for 10 min on ice bath with Qsonica MC-18 sonicator, the supernatant was isolated by centrifugation at 29,097 × g for 45 min at 4 °C. Clear supernatants were applied on a nickel-nitrilotriacetic (Ni-NTA) agarose resin (Qiagen, ref. 30250) compacted in gravity columns and eluted with a buffer containing 50 mM Tris–HCl pH 7.5, 1 M NaCl, 500 mM imidazole, 2 mM TCEP, 0.04% NaN$_3$. Eluted proteins were cleaved with TEV protease overnight at 4 °C, in a buffer containing 50 mM Tris–HCl pH 7.5, 150 mM NaCl, 20 mM imidazole, 2 mM TCEP, and 0.04% NaN$_3$. The supernatant of SRPK1 was applied to Cytiva HisTrap hp 5 mL column and has been exchanged to the phosphorylation buffer containing 50 mM Tris-HCl pH 6.7, 150 mM NaCl, 20 mM MgCl$_2$, 2 mM TCEP, and 0.04% NaN$_3$ immediately after elution without TEV cleavage. Untagged proteins were further isolated using a second affinity purification gravity column with Ni-NTA beads. A final size exclusion chromatography purification step was performed in the buffer of interest on a gel filtration column (Superdex 30, GE Healthcare for CIRBP$^{RSY}$ and Hiload 16/600 Superdex 200 pg, GE Healthcare for TNPO3). Untagged SRSF1$^{RS}$ has been isolated using a HiTrap Heparin HP affinity column (Cytiva) and eluted with a buffer containing 50 mM Tris, pH 7.5, and 1 M NaCl followed by desalt into the buffer of interest.

Protein concentrations were estimated according to their absorbance at 280 nm, assuming that the $\varepsilon$ at 280 nm was equal to the theoretical $\varepsilon$ value.

### Synthetic peptides
Synthetic peptides corresponding to CIRBP (residues 157–172; wild-type and mutants R161A, Y164A, Y167A), phosphorylated variants (Y164pY, S166pS, Y167pY), as well as peptides from P5CS (3–20), RBMX (276–291), RBM39 (86–101), and ZCCHC18 (371–390), were purchased from Peptide Specialty Laboratories GmbH (Heidelberg, Germany).

### Isothermal titration calorimetry (ITC)
ITC has been performed by a Malvern Microcal VP-ITC at 25 °C for all binding studies. Samples have been dissolved in Tris buffer containing 50 mM Tris–HCl, 150 mM NaCl, 2 mM TCEP, and 0.04% NaN3 at pH 7.5. Binding studies of CIRBP$^{RSY}$ and SRSF1$^{RS}$ were performed with TNPO3 WT, and studies of CIRBP (157–172) and its mutants/phosphorylations were performed with TNPO3 C511A, respectively. Injection has been carried out by a syringe containing ligand solution of 100 µM, which adds ligand step-wise into the cell containing 10 µM TNPO3 during 28 injections of 10 µL each. Data analysis, curve fitting, and calculation of $K_d$, $\Delta H$, and $\Delta S$ have been carried out using Origin-based 7.0 software provided by the manufacturer. All replicated ITC data are shown in Supplementary Fig. 6 and Supplementary Table 1.

### In vitro phosphorylation
The phosphorylation procedure used in this study followed the protocol established in the previous report[42]. CIRBP$^{RSY}$ (or SRSF1$^{RS}$) and SRPK1 were equilibrated in phosphorylation buffer containing 50 mM Tris, pH 6.7, 150 mM NaCl, 20 mM MgCl$_2$, 2 mM TCEP, and 0.04% NaN$_3$, 10% D$_2$O was included only when the reaction mixture was also used as an NMR sample for kinetic measurements. Immediately preceding the reaction, solutions containing the peptide and SRPK1 were mixed at a stoichiometric ratio of 2:1 (peptide:SRPK1). Additional 10 mM ATP has been added to trigger the reaction overnight. To isolate the phosphorylated peptide and exchange it into the desired buffer, the reaction mixture has been concentrated by Millipore Amicon Ultra-15

Centrifugal Filter (PLBC, membrane Ultracel-PL, 3 kD, UFC9003) followed by a size exclusion chromatography using a Superdex™ 30 Increase column (Cytiva).

## NMR spectroscopy

All NMR studies for TNPO3 were carried out using the wild type protein. Protein samples for $^1$H-$^{13}$C HMQC were prepared at a concentration of 100 μM and equilibrated in a buffer containing 50 mM Tris−HCl, pH 6.7, 150 mM NaCl, 2 mM TCEP and 10% $D_2O$. Spectral acquisitions for TNPO3 free and in complex with CIRBP$^{RSY}$ and SRSF1$^{RS}$, respectively were performed at 298 K on a 700 MHz Bruker Avance III NMR spectrometer equipped with a TCI triple-resonance cryoprobe. The pulse program was sfhmqcf2gpph and the spectra were recorded with 32 scans, interscan delay of 1 s, 160 points in F1, 1024 points in F2, 5283.879 Hz spectral width in F1, and 11,160.714 Hz spectral width in F2. Spectra were processed using Topspin 4.1.3.

Spectral acquisitions for TNPO3 free and in complex with mCIRBP$^{RSY}$, pCIRBP$^{RSY}$, and pSRSF1$^{RS}$, respectively, were performed at 298 K on a 600 MHz Bruker Avance Neo NMR spectrometer equipped with a TXI 600S3 probehead. The pulse program was hmqcgpphwg and the spectra were recorded with 64 scans, interscan delay of 1 s, 200 points in F1, 1024 points in F2, 4545.455 Hz spectral width in F1, and 7812.500 Hz spectral width in F2. Spectra were processed using Topspin 4.3.0.

For $^1$H-$^{15}$N HSQC spectra, protein samples were equilibrated in a buffer containing 50 mM Tris−HCl, pH 6.7, 150 mM NaCl, 2 mM TCEP, 0.04% NaN$_3$, and 10% $D_2O$ (with additional 20 mM MgCl$_2$ for CIRBP phosphorylation kinetics). Data were recorded at 298 K on the 600 MHz Bruker Avance Neo NMR spectrometer equipped with a TXI 600S3 probehead. The pulse program was hsqcetfpf3gpsi and was recorded with 8-16 scans, interscan delay of 1 s, 128 points in F1, 1024 points in F2, 1949.318 Hz spectral width in F1, and 9615.385 Hz spectral width in F2. Spectra were processed using Topspin 4.3.0.

Most spectra were directly exported for figure preparation using Topspin 4.3.0 or UCSF Sparky (T. D. Goddard and D. G. Kneller, SPARKY 3, University of California, San Francisco). The assignment of phosphorylated CIRBP$^{RSY}$ was transferred from the previously published dataset (BMRB ID: 28025 and ref. 30) and data was analyzed by CCPNmr 2.5.2[57], peak intensity over time was plotted using GraphPad Prism 9. In the previous study, the unphosphorylated CIRBP$^{RSY}$ was assigned under buffer conditions containing 50 mM Tris, 150 mM NaCl, 2 mM TCEP, 0.04% NaN$_3$ and 10% $D_2O$, and adjusted to pH 6.7. The same buffer conditions were used in the present study, with the addition of 20 mM MgCl$_2$, which did not affect the backbone chemical shift of CIRBP$^{RSY}$.

## Crystallization, data acquisition and processing

All crystallization experiments were performed using sitting drop vapor diffusion in a 96-well 3-drop plate (SwissCI AG, Neuheim, Switzerland) and the ORYX8 pipetting robot (Douglas Instruments, Hungerford, UK). Commercial screens have been used for initial screening and optimization: Index HR2-144 (Hampton Research, United States), JCSG+, SG1, and Wizard Classic 1-4, and Morpheus I, II, III (Molecular Dimensions, United States). Purified TNPO3 C511A with 12 mg/mL and CIRBP with 4.5 mg/mL were used for crystallization. Each drop was set up with 0.5 μL protein mix with a 1:4 molar ratio TNPO3: CIRBP and 0.5 μL screening solution. Additional 0.1 μL of a 10-7 diluted seeding stock from previous experiments was added. Crystallization plates were incubated at 4 °C and the first crystals grew after 10 days.

Conditions with high PEG concentrations yielded the most crystals, whereas condition E10 from the Morpheus III screen (Buffer: 0.1 M Sodium HEPES, 0.1 M MOPS (acid), 7.5 pH, Precipitant: 12.5% v/v MPD; 12.5% PEG 1000; 12.5% w/v PEG 3350, Additives: 0.05% w/v D-Salicin, 0.05% w/v Esculin hydrate, 0.05% w/v Quinine hemisulfate salt monohydrate, 0.05% w/v Tryptamine, 0.05% w/v Arbutin (in 50%

EtOH)) resulted in the best diffracting crystal. The crystals were rather thin, square shaped 150 μm in diameter layered plates.

Data collection was performed at the synchrotron-radiation facility DESY PETRA III, Beamline P11 (Hamburg, Germany) on 2021-02-05 with a DECTRIS EIGER2 X 16 M detector at 100 °K and a wavelength of 1.03322 Å. A total of 1800 images with 0.2° oscillation range, 10 ms exposure time, 50 μm focus size and 100 μm pinhole size were collected. For further processing only the first 900 images were used due to heavy radiation damage and smearing of spots in the second half of the dataset.

Raw data was processed using XDS (20230630)[58] for indexing and integrating and data reduction was done with AIMLESS (0.7.7)[59]. The best crystal diffracted up to 2.95 Å with lattice constants of 97.5 Å 101.8 Å 114.1 Å and 90° 111.1° 90° and P2$_1$ symmetry. The structure was solved with Phaser (2.8.3)[60] and molecular replacement with 4C0P (unliganded TNPO3) as template. The model was prior to MR cut into several smaller domains as phasing with the whole structure did not result in a proper solution. There are two chains of TNPO3 in the ASU whereas chain B shows large flexible parts near the N-terminal end. After rebuilding and improving the structure with Coot (v.0.9.8.93)[61] and refinement with REFMAC5[62] there was some clear density for CIRBP bound to each chain and the peptides were built. Additional there is some density for a weak bound third molecule of CIRBP bound to Chain B, facing the solvent. Water molecules, clashes, Ramachandran and rotamer outliers were improved within ChimeraX (v.1.7)[63] and the ISOLDE Plugin (v.1.6)[64]. The final refinement statistics were generated with Phenix (1.20.1)[65] and are shown in Table 2. The structure is deposited at the PDB RSCB databank and has the accession code 8CMK. The omit map of CIRBP and electron density around the binding site have been shown in the Supplementary Fig. 7.

## Hormone-induced nuclear import assay

To analyze import of GCR$_2$-GFP$_2$ tagged CIRBP reporters, experiments were performed as described previously[40]. In brief, HeLa cells (provided by Marc-David Ruepp, KCL London) were grown for at least 2 passages in DMEM supplemented with 10% dialyzed FBS and transiently transfected with GCR$_2$-GFP$_2$-CIRBP wildtype (WT), R161A or Y164A mutant seeded in ibidi 8-well dishes using lipofectamine 2000. A GCR$_2$-GFP$_2$-stop reporter served as control for entry into the nucleus by diffusion. Import of the GCR$_2$-GFP$_2$-reporter was induced by adding dexamethasone (Sigma D4902, 5 μM final concentration) in imaging medium (DMEM fluorobrite) and followed over time by live cell imaging. Images were acquired for a duration of 30 min in 2.5 min intervals in an environmental chamber at 37 °C and 5% $CO_2$ using an inverted spinning disc microscope (Visiscope 5 Elements, Visitron Systems GmbH, Puchheim, Germany), built on a Nikon Ti2 stand equipped with a confocal spinning disc (CSU-W1; Yokogawa, Tokyo, Japan) with 50 μm pinhole diameter and a 60x/1.2 NA water immersion objective. Imaging was performed using the 488 nm laser line for excitation and 565/133 bandpass filter (Semrock) for collecting the fluorescence emission. Images were acquired using sCMOS camera (Prime BSI, Photometrics) with a full frame (2048 × 2048 pixels) and 12-bit acquisition mode. Analysis was performed using Fiji/Image J (1.54). If applicable, the StackReg Plugin was used to compensate xy drift of cells over time before downstream analysis. To determine nuclear import, at least 40 cells per replicate were analyzed by measuring a representative circular region of interest in nucleus (N) and cytoplasm (C), respectively, and background subtracted values plotted as N/C ratio over time using GraphPad Prism 9.

## Bioinformatic motif scan

Scan of motifs in the human proteome has been carried out with the ScanProsite tool available through the Expasy PROSITE (https://

**Table 2 | Data collection and refinement statistics of the crystal strucutre of TNPO3 with bound ligand CIRBP (8CMK)**

| | 8CMK |
|---|---|
| *Data collection* | |
| Wavelength (Å) | 1.0332 |
| Space group | P 2₁ |
| Cell dimensions | |
| a, b, c (Å) | 97.53, 101.81, 114.13 |
| α, β, γ (°) | 90, 111.13, 90 |
| Resolution (Å) | 48.72–2.945 (3.05–2.945) |
| $R_{merge}$ | 0.06095 (0.5516) |
| $I/\sigma I$ | 8.07 (1.58) |
| Completeness (%) | 98.26 (95.79) |
| Redundancy | 2.0 (1.9) |
| CC ½ | 0.997 (0.466) |
| *Refinement* | |
| No. of reflections | 43,516 (4211) |
| $R_{work}$ | 0.2484 (0.3314) |
| $R_{free}$ | 0.3165 (0.4047) |
| No. of atoms | 14,823 |
| Protein | 14,632 |
| Ligand/ion | 19 |
| Water | 172 |
| *B*-factors | 85.58 |
| Protein | 86.01 |
| Ligand/ion | 74.26 |
| Water | 50.58 |
| R.m.s. deviations | |
| Bond lengths (Å) | 0.008 |
| Bond angles (°) | 1.56 |
| Ramachandran | |
| Favored (%) | 97.48 |
| Allowed (%) | 2.41 |
| Outliers (%) | 0.11 |
| Rotamer outliers (%) | 0.49 |
| Clashscore | 2.12 |

Statistics for the highest resolution shell are given in parentheses.

prosite.expasy.org/scanprosite/, SIB Swiss Institute of Bioinformatics) webserver[38]. The respective motifs were queried against the UniProtKB database and filtered by the taxonomy *Homo sapiens* (9606). For proteins with consecutive SR/RS motifs, we searched all combinations of (S-R/R-S)-(S-R/R-S)-(S-R/R-S), i.e. S-R-S-R-S-R, R-S-S-R-S-R, S-R-R-S-S-R, etc, and downloaded the result of all queries in the Supplementary Data 1.

Additional scan of motifs was conducted for the motif Y-R-x(2,3)-Y-x(2,3)-Y and [YWF]-R-x(2,3)-[YWF]-x(2,3)-[YWF]. The results of these scans, along with further manual selections, are included in the Supplementary Data 2 and 3, respectively.

For all proteins with motif Y-R-x(2,3)-Y-x(2,3)-Y (Supplementary Data 2) and the 12 proteins containing [YWF]-R-x(2,3)-[YWF]-x(2,3)-[YWF] (Supplementary Fig. 5 and Supplementary Data 3, Datasheet 2), secondary structure examination of the motifs was conducted using AlphaFold2-predicted structures available via UniProt[39,66,67]. The identifiers of AlphaFold2-predicted structures are included in the Supplementary Data 2 and 3 with all structures accessed in July 2023.

The list of human nuclear proteins (ID SL-0191, GO:0005634, *Homo sapiens*, and reviewed) dated to July 2024 was downloaded from UniProt database (https://www.uniprot.org/)[39], and has been shown as Supplementary Data 4.

### AlphaFold2 prediction
Predictions of TNPO3-ligand binding complex, and predictions of protein structures containing putative binding motifs have been carried out by AlphaFold2 using MMseqs2 (i.e., AlphaFold2-multimer model) that has been included in ColabFold - v1.5.5[66,68] and using the webserver: https://colab.research.google.com/github/sokrypton/ColabFold/blob/main/AlphaFold2.ipynb. The input consists of the whole protein sequence of TNPO3 and the sequence of the ligand, being separated into two chains with number of relax at five and no template mode. Calculations were carried out on the GPU provided by Google-Colab, all additional parameters have been selected as default (msa_mode: mmseqs2_uniref_env; pair_mode: unpaired_paired; model_type: auto; num_recycles:3; recycle_early_stop_tolerance: auto; relax_max_iterations: 200; pairing_strategy: greedy; max_msa: auto; num_seeds: 1. The raw output data generated by AlphaFold2 have been deposited in the Zenodo repository (https://zenodo.org, https://doi.org/10.5281/zenodo.15268248).

### Reporting summary
Further information on research design is available in the Nature Portfolio Reporting Summary linked to this article.

## Data availability
The structure of the TNPO3-CIRBP^RSY complex generated in this study was deposited in the RCSB Protein Data Bank under the accession code 8CMK. PDB codes of previously published structures used in this study are 4C0O (Transportin 3 in complex with phosphorylated ASF/SF2), 6GX9 (TNPO3 - CPSF6 RSLD complex), and 4C0P (Unliganded Transportin 3). NMR chemical shift assignments were transferred from previously published data BMRB 28025 (human_CIRBP_138-172). Raw NMR data can be reproduced based on the protein preparation details in the "Methods" section. AlphaFold2 structures generated in this study have been deposited in the Zenodo repository (https://zenodo.org, https://doi.org/10.5281/zenodo.15268248). Researchers seeking access to original datasets not publicly available may contact the corresponding author. Source data are provided with this paper.

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

## Acknowledgements

T.M. is grateful to the Austrian Science Fund (FWF) for excellence cluster 10.55776/COE14, Grants DOI 10.55776/P28854, 10.55776/I3792, 10.55776/DOC130, and 10.55776/W1226, the Austrian Research Promotion Agency (FFG) grants 864690 and 870454; the Integrative Metabolism Research Center Graz; the Austrian Infrastructure Program 2016/2017; the Styrian Government (Zukunftsfonds, doc.fund program); the City of Graz; and BioTechMed-Graz (flagship project). This project was funded in part by the FFG (https://www.ffg.at/) and the European Union (EFRE) under grant 912192. The work was supported by the Austrian Science Fund (FWF) (Grant DOI and 10.55776/P29432 to T.P.-K.); as well as the Deutsche Forschungsgemeinschaft (DFG) (project number 397983986 and the Heisenberg program (project number 442698351) to D.D.). Q.Z. was trained within the frame of the PhD programs DK-MCD and BioMolStruct, and T.S. was trained within the frame of the PhD program MOBILES. The Spinning Disk Confocal System (VisiScope 5-Elements, IMB Microscopy Core Facility) was supported by the Deutsche Forschungsgemeinschaft (INST 247/912-1FUGG). We are grateful for the IMB Microscopy facility staff for technical support. We thank the Center for Medical Research, Medical University of Graz, and Institute of Molecular Biosciences, University of Graz, Graz, Austria for laboratory access. We thank Marc-David Ruepp for providing the HeLa cells. For open access purposes, the author has applied a CC BY public copyright license to any author accepted manuscript version arising from this submission.

## Author contributions

B.B. and T.M. designed the study; Q.Z. performed recombinant protein acquisitions, in vitro phosphorylation, bioinformatic studies, AlphaFold2 prediction, and wrote the manuscript; Q.Z., B.B., and T.M. carried out NMR experiments; T.S. performed crystallography; S.H. performed cell-based studies; Q.Z. and B.B. performed ITC studies; Q.Z., T.S., B.B., S.H., and T.M. performed data analysis and interpretation; Q.Z., T.S., B.B., S.H., T.P.-K., T.M., and D.D. performed corrections on the manuscript; T.M. and D.D. acquired fundings.

## Competing interests

The authors declare no competing interests.
