## [Transparent Peer Review file · Nature Communications]

Structural basis of phosphorylation-independent nuclear import of CIRBP by TNPO3

Corresponding Author: Professor Tobias Madl

Version 0:

Reviewer comments:

Reviewer #1

(Remarks to the Author)

TNPO3 is an important karyopherin involved in some human pathologies and HIV-1 infection. Its best characterized biological function is in nuclear import of SR and SR-like repeat containing proteins, such as SFRS1 and CPSF6.

In their study, Madl and co-workers show that TNPO3 also acts as a major nuclear import factor for proteins containing an alternative NLS containing aromatic and Arg residues: YRxYxY. Intriguingly, TNPO3 engages these alternative NLSs via the same region, which it uses to recognize phosphorylated SR/RS repeats. Yet, by contrast to the classical SR NLSs, the alternative NLSs do not require phosphorylation. The study nicely explains the structural basis for a remarkable adaptation of the karyopherin to a different type of cargo. This is an interesting and important work, although some issues need to be addressed before publication.

Major:

1. The study is based on a previous work (reference #30) that implicated two karyopherins (TNPO1 and TNPO3) in nuclear import of CIRBP. Yet, TNPO1 seems to be completely ignored throughout the manuscript. Knockdown of endogenous TNPO3 and supplementation with WT and mutant TNPO3 (using mutants based on the co-crystal structure) is required for functional validation.

Minor:

2. A couple of residues is not an alpha helix; the residues must form at least one full turn to be a helix.

3. Indicate R helix in the figures; this element in TNPO3 structure is central for the "classical" TNPO3 pSR/RpS NLS recognition and is clearly involved in the YR NLS binding.

Reviewer #2

(Remarks to the Author)

This manuscript reports the detailed structural and functional characterisation of a previously unknown motif binding to transportin 3 (TNPO3). This motif is distinct from the canonical NLS motif reported to bind TNPO3, and is regulated in an opposite manner by phosphorylation. This study is therefore novel and of interest to a wide audience. However, I have some comments on the different structural biology experiments performed to characterize the new motif, and in particular, I feel that the sentence « we demonstrated through NMR spectroscopy and AlphaFold2 predictions that the RBMX/RMXL1 motifs directly interact with TNPO3 » is not strongly supported.

Figure 1: why were the NMR spectra of panels B and C recorded at 298 K? At this temperature, the serine peaks are difficult to detect, and I cannot see peaks corresponding to each residue of the peptide. Why didn't the authors use a temperature of

283 K? Also, in Mat & Met it is written that the NMR experiments were based on a HMQC scheme, and in the legend it is written HSQC. More generally, the Mat & Met section entitled NMR spectroscopy is a bit confusing: where are the parameters of the ¹H-¹⁵N experiments?

About the ITC experiments: I did not find the corresponding Mat & Met section. Also, could the authors show a duplicate of the ITC experiments of panel D?

Figure 2: Is TNPO3 a monomer in solution? In order for the reader to understand where are the additional contact points described in Sup Fig 2, a global view of the different contact points would be helpful.

Minor points: why is the panel A called « Zoom », as we see the whole structure, correct? Typo: « the contact map have >> has been generated ... ». What does helix 14-17 mean (I can understand it but it is still an unusual way to name a series of helices)?

Figure 3: could the PDB codes of the crystal structures be cited? Can the authors comment the differences of affinities measured between TNPO3 and either CIRBP(137-172) or CIRBP(157-172)? Can they show replicated ITC experiments supporting the main ITC results? Can they comment which residue is found at the position of the phospho-residue in the canonical complexes (this residue is only briefly cited in lines 193-195)?

Figure 4: How were the proteins listed in panel A selected from the set of 90 proteins cited in the Results section? The residue numbers of CIRBP seem not to correspond to the sequence. Could panel D be added to the Suppl Data and Figures 4 and 5 assembled into a single figure?

Figure 5: It is not so easy to identify the NMR peaks of the ligand in the 1D spectra of the complexes (in particular in the case of mP5CS). So that the concentration measurements have to be reliable to interpret the loss of ligand peak intensities as resulting from binding. Couldn't the authors perform an STD experiment to clearly detect binding? A confirmation of binding by another method could also be useful, and could tell if there are large affinity differences between the ligands.

About AlphaFold: could the authors give some metrics corresponding to the AlphaFold results (ipTM scores, IDDT plots), supporting the models? To have an idea of the degree of confidence of the predictions? If the scores are really too low, then AlphaFold does not demonstrate anything.

Figure 6: Why is it relevant to monitor phosphorylation of CIRBP by SRPK1? Could the authors highlight on the CIRBP sequence which residues are phosphorylated by SRPK1 and which residues are involved in TNPO3 binding? Do the phosphorylated residues correspond to canonical SRPK1 phosphosites? How did the authors estimate the intensities of the NMR peaks corresponding to totally (100%) phosphorylated residues?

Also, can the authors comment, based on their crystal structure, why phosphorylation of so many residues impairs binding to TNPO3, whereas it favors binding in the case of the canonical motif?

Version 1:

Reviewer comments:

Reviewer #1

(Remarks to the Author)

My concerns have been adequately addressed in revision.

Reviewer #2

(Remarks to the Author)

The authors have carefully answered to my questions.

REVIEWER COMMENTS

Comments are highlighted with a green background

Reviewer #1 (Remarks to the Author)

TNPO3 is an important karyopherin involved in some human pathologies and HIV-1 infection. Its best characterized biological function is in nuclear import of SR and SR-like repeat containing proteins, such as SFRS1 and CPSF6.

In their study, Madl and co-workers show that TNPO3 also acts as a major nuclear import factor for proteins containing an alternative NLS containing aromatic and Arg residues: YR_xY_xY. Intriguingly, TNPO3 engages these alternative NLSs via the same region, which it uses to recognize phosphorylated SR/RS repeats. Yet, by contrast to the classical SR NLSs, the alternative NLSs do not require phosphorylation. The study nicely explains the structural basis for a remarkable adaptation of the karyopherin to a different type of cargo. This is an interesting and important work, although some issues need to be addressed before publication.

We thank the reviewer for positive evaluation and her/his suggestions and conducted the additional analyses/experiments requested.

The study is based on a previous work (reference #30) that implicated two karyopherins (TNPO1 and TNPO3) in nuclear import of CIRBP. Yet, TNPO1 seems to be completely ignored throughout the manuscript. Knockdown of endogenous TNPO3 and supplementation with WT and mutant TNPO3 (using mutants based on the co-crystal structure) is required for functional validation.

In our previous study (Bourgeois et al., PMID: 32234784), we demonstrated that CIRBP contains two distinct nuclear localization signals (NLSs): the RG/RGG region, primarily imported via TNPO1, and the RSY region, predominantly recognized by TNPO3. Functional validation in that study included the deletion of the CIRBP RSY region and TNPO3 knockdown (see ptp_reply_Figure 1, panels D and E), both of which significantly impaired CIRBP nuclear import, highlighting the crucial role of the RSY region. However, the precise mechanism underlying TNPO3 recognition of the RSY region remained unclear.

We also found that TNPO3 binds CIRBP's RSY region with ~70-fold higher affinity than TNPO1. Given this, our current study focuses on the structural basis of TNPO3-mediated nuclear import via the RSY-NLS. The role of TNPO1, particularly in recognizing non-canonical cargoes such as CIRBP, will be explored in future work. We have added a statement related to TNPO1 on page 4:

“Notably, while this RSY-NLS is also recognized by transportin-1 (TNPO1), its binding affinity is approximately 70 times lower than that of TNPO3.”

[REDACTED]

ptp_reply_Figure 1. Taken from Bourgeois et al., PMID: 32234784. [REDACTED]

A couple of residues is not an alpha helix; the residues must form at least one full turn to be a helix.

We apologize for the confusion raised and corrected the statements on page 6 and 7 accordingly:

“For the CIRBP^{RSY} peptide, only 14 residues (amino acids 158–171) from the full CIRBP fragment (amino acids 138–172) could be modeled unambiguously. These residues form two short helices, each containing a single turn (residues 159–163 and 166–170; Figure 2B).”

“Unlike the canonical ligands pSRSF1 and pCPSF6, which lacks a helix at the binding site, CIRBPRSY forms two short helices between S159-S163 and S166-H170, respectively (Figure 3A/B).”

Indicate R helix in the figures; this element in TNPO3 structure is central for the "classical" TNPO3 pSR/RpS NLS recognition and is clearly involved in the YR NLS binding.

Thank you for your suggestion, we have highlighted the R helix in Figure 2B, Figure 3A, and Figure 3B.

Reviewer #2 (Remarks to the Author)

This manuscript reports the detailed structural and functional characterisation of a previously unknown motif binding to transportin 3 (TNPO3). This motif is distinct from the canonical NLS motif reported to bind TNPO3, and is regulated in an opposite manner by phosphorylation. This study is therefore novel and of interest to a wide audience. However, I have some comments on the different structural biology experiments performed to characterize the new motif, and in particular, I feel that the sentence « we demonstrated through NMR spectroscopy and AlphaFold2 predictions that the RBMX/RMXL1 motifs directly interact with TNPO3 » is not strongly supported.

We thank the reviewer for positive evaluation and her/his suggestions.

Figure 1: why were the NMR spectra of panels B and C recorded at 298 K? At this temperature, the serine peaks are difficult to detect, and I cannot see peaks corresponding to each residue of the peptide. Why didn't the authors use a temperature of 283 K? Also, in Mat & Met it is written that the NMR experiments were based on a HMQC scheme, and in the legend it is written HSQC. More generally, the Mat & Met section entitled NMR spectroscopy is a bit confusing: where are the parameters of the ^1H - ^{15}N experiments?

We apologize for the confusion raised and thank the reviewer for her/his suggestions.

The purpose of this section was to first validate that phosphorylation of the RS region of SRSF1 enhances TNPO3 binding, as previously demonstrated by others. Our NMR and ITC experiments strongly support this conclusion. The NMR experiments were conducted primarily to verify the efficiency of our phosphorylation protocol. Specifically, we observed the appearance of multiple ^1H - ^{15}N HSQC cross-peaks in the region characteristic of phosphorylated serines, along with the corresponding disappearance of non-phosphorylated peaks.

Regarding Figure 1, panels B and C, our goal was to compare TNPO3 binding to the unphosphorylated and phosphorylated SR/RS motif of SRSF1. Given the clear distinction between these states, we did not assign individual chemical shifts for the resonances.

As for the type of 2D NMR spectra, we used:

- ^1H , ^{13}C HMQC for [IM-methyl ^{13}C]-labeled TNPO3 in both its bound and unbound states
- ^1H , ^{15}N HSQC for ^{15}N -labeled ligands (SRSF1^{RS} and CIRBP^{RSY})

We have now updated the figure legends and Materials & Methods section to correct any inconsistencies and ensure clarity, including the parameters of the ^1H - ^{15}N experiments.

About the ITC experiments: I did not find the corresponding Mat & Met section. Also, could the authors show a duplicate of the ITC experiments of panel D?

We have added a description of the ITC experiment in the Method section. Please find the replicates corresponding to the ITC experiments related to panel D below, as well as for the mutant and phosphorylated peptides. In summary, the results of the replicates are in excellent agreement with the data shown previously. We have included the additional ITC data as new Supplementary Figure S6 and added the results to Table S1. This is also indicated at the end of ITC section in the Materials and Methods.

ptp_reply_Figure 2. Replicate ITC data for TNPO3 WT/C511A binding to target peptides. (A-B) ITC curve showing titrations of 100 μM CIRBP^{RSY} into 10 μM of (A) TNPO3 and (B) TNPO3 C511A mutant. (C) ITC curve showing titration of 100 μM phosphorylated pCIRBP^{RSY} into 10 μM of TNPO3. (D) ITC curve showing titrations of 100 μM SRSF1^{RS} (left panel) and phosphorylated pSRSF1^{RS} (right panel) into 10 μM TNPO3 WT. (E) ITC curve of titrations of 100 μM peptides derived from CIRBP^{RSY} with alanine mutations into 10 μM of TNPO3 C511A. This includes mCIRBP^{RSY} (157-172, left panel 1), R161A mutant of mCIRBP^{RSY} (157-172, left panel 2), Y164A mutant of mCIRBP^{RSY} (157-172, left panel 3) and Y167A mutant of mCIRBP^{RSY} (157-172, right panel). (F) ITC curve of titrations of 100 μM peptides derived from CIRBP^{RSY} with phosphorylations into 10 μM of TNPO3 C511A. This includes mCIRBP^{RSY} (157-172, left panel 1), Y164pY mutant of mCIRBP^{RSY} (157-172, left panel 2), S166pS mutant of mCIRBP^{RSY} (157-172, left panel 3) and Y167pY mutant of mCIRBP^{RSY} (157-172, right panel). The reported errors correspond to the SD of the fit.

Figure 2: Is TNPO3 a monomer in solution? In order for the reader to understand where are the additional contact points described in Sup Fig 2, a global view of the different contact points would be helpful.

TNPO3 has been previously shown by SAXS to exist in a monomer/dimer equilibrium in solution with an associated K_D of $8.3 \pm 1.0 \mu\text{M}$ (R Larue *et al.* 2012, PMID: 22872640). Out of the additional contact points in the crystal structure, the intermolecular salt bridge of E304 with K877 is the only contact which we could observe in both directions (chain A – chain B and *vice versa*). We included an additional figure to the supplement (Figure S2C) in which we show this contact. Moreover, we have added a short statement referring to the previous work:

“Among the intermolecular dimer contacts observed between TNPO3 molecules in chains A and B, the only reproducible interaction was a potential salt bridge between E304 and K877 (Figure S2C and S2D). Notably, TNPO3 dimerization was also observed in solution, with a dissociation constant of $8.3 \pm 1.0 \mu\text{M}$. However, since endogenous TNPO3 concentrations are likely below this threshold, dimer formation is unlikely to be physiologically relevant under normal cellular conditions.”

Minor points: why is the panel A called « Zoom », as we see the whole structure, correct? Typo: « the contact map have >> has been generated ... ». What does helix 14-17 mean (I can understand it but it is still an unusual way to name a series of helices)?

We deleted “zoom” from the corresponding Figure legend and corrected the typo. We modified the expression “helix 14-17” in the introduction and result section to “between HEAT repeats 14 and 17”.

Figure 3: could the PDB codes of the crystal structures be cited? Can the authors comment the differences of affinities measured between TNPO3 and either CIRBP(137-172) or CIRBP(157-172)? Can they show replicated ITC experiments supporting the main ITC results? Can they comment which residue is found at the position of the phospho-residue in the canonical complexes (this residue is only briefly cited in lines 193-195)?

We added PDB codes and associated citations for all used 3D structures throughout the paper.

We have previously shown using NMR titrations that residues around CIRBP Y141 display line broadening upon addition of TNPO3 (see *ptp_reply_Figure 3*). However, as these residues were not resolved in the crystal structure, we refrained from further interpretations.

[REDACTED]

ptp_reply_Figure 3. Distinct CIRBP regions are directly recognized by TNPO1 and TNPO3. Figure taken from (Bo urgeois B. et al. 2020, PMID: 32234784, Figure 3D). [REDACTED]

We performed replicate ITC experiments and have included the results in the revised manuscript as Supplementary Figure S6 and Table S1. All replicates show strong consistency with the previously presented data.

Regarding the position of phospho-residues in the canonical complexes, we have added a clarifying statement to the revised manuscript:

"When comparing the positions of the phospho-residues in the canonical complexes to those of CIRBP residues, we observed no direct structural overlap. For instance, CIRBP Y160 and A168 are located near SRSF1 pS207/pS209, while CIRBP R161 aligns closely with CPSF6 pS525."

Figure 4: How were the proteins listed in panel A selected from the set of 90 proteins cited in the Results section? The residue numbers of CIRBP seem not to correspond to the sequence. Could panel D be added to the Suppl Data and Figures 4 and 5 assembled into a single figure?

The proteins listed in panel A are the fraction of the 90 proteins that possess the Y-R-x(2,3)-Y-x(2,3)-Y motif in disordered regions, as predicted by AlphaFold 2. Proteins in which the motif is located in well-structured regions of the AlphaFold 2 models were excluded. To avoid confusion, we now first discuss the results of the AlphaFold 2 calculations and show the peptides which have been selected for further binding studies (see also other ptp replies below):

“This analysis identified over 90 proteins, 33 of which contain the motif within their IDRs (Supplementary Datasheet S2).”

As suggested by the reviewer we have reorganized the data and merged Figures 4 and 5.

Figure 5: It is not so easy to identify the NMR peaks of the ligand in the 1D spectra of the complexes (in particular in the case of mP5CS). So that the concentration measurements have to be reliable to interpret the loss of ligand peak intensities as resulting from binding. Couldn't the authors perform an STD experiment to clearly detect binding? A confirmation of binding by another method could also be useful, and could tell if there are large affinity differences between the ligands.

We changed the spectra displayed in Figure 4 (former Figure 5) to better visualize the extensive line-broadening of the RSY peptide aromatic proton resonances regions upon addition of 1 stoichiometric equivalent of TNPO3.

Unfortunately, using STD-NMR experiments using 100 μM of RSY regions and 10 μM of TNPO3 we could not detect a STD effect, likely because the exchange regime associated to these complex formation is not compatible with STD detection.

Nevertheless, we also performed ^1H 1D NMR with these samples and confirmed in another experimental setup (TNPO3 associated signals are almost invisible) that all aromatic protons from the tested RSY regions become extensively broadened in the presence of TNPO3 (see below).

ptp_reply_Figure 4. (A) Overlay of ^1H 1D NMR spectra of potential TNPO3 binding peptides free (black), and in presence of a sub-stoichiometric amount of TNPO3 (red). For the peptides binding with high affinity (slow exchange on the NMR time scale), such as CIRBP^{RSY} and RBMX^{RSY} a slight decrease of peptide signal intensity was observed. For the other peptides, line broadening of ^1H resonances were observed upon addition of substoichiometric amounts of TNPO3.

About AlphaFold: could the authors give some metrics corresponding to the AlphaFold results (ipTM scores, IDDT plots), supporting the models? To have an idea of the degree of confidence of the predictions? If the scores are really too low, then AlphaFold does not demonstrate anything.

Thank you for your suggestions. We have added the pTM/ipTM values to **Table S2**. The IDDT scores were generally high for TNPO3 (IDDT ~90), but lower for the peptides (IDDT 20-70). For CIRBP^{RSY}, AlphaFold2 correctly predicted the global localization of the binding site with an IDDT score of around 30, although with an incorrect local structure.

We have added the following statement to the revised version of the manuscript:

“While these known NLSs yielded relatively low IDDT scores, AlphaFold2 correctly predicted their binding locations.”

ptp_reply_Figure 5. Predicted aligned error (PAE, showing the rank 1-3 of prediction) and predicted local distance difference test (PLDDT) plots matching the AlphaFold2 prediction of TNPO3 binding to the RSY regions of (A) CIRBP, (B) P5CS, (C) RBM39, (D) ZCH18, and (E) RBMX. To be noted that 1) the putative binding site of ligand is associated to a slightly lower PAE, as indicated by the arrow, and 2) prediction of TNPO3 structure is associated to a high LDDT value, whereas only peptide RBMX yielded a value around 60.

Figure 6: Why is it relevant to monitor phosphorylation of CIRBP by SRPK1? Could the authors highlight on the CIRBP sequence which residues are phosphorylated by SRPK1 and which residues are involved in TNPO3 binding? Do the phosphorylated residues correspond to canonical SRPK1 phosphosites? How did the authors estimate the intensities of the NMR peaks corresponding to totally (100%) phosphorylated residues?

SRPK1 is involved in the phosphorylation of several proteins harboring RS/SR di-repeat motifs and therefore regulates their binding to TNPO3 and nuclear import. We previously showed that SRPK1 could phosphorylate the RG/RGG region of CIRBP and modulates TNPO1 binding. Here, we aimed to elucidate whether it could also modulate TNPO3 binding via phosphorylation of the RSY region. We could assign three SRPK1-associated phosphosites in the RSY region, S146, S148 and S152 which are all located in between the two RSY sub-regions involved in TNPO3 binding (according to NMR-derived CSPs, see ptp_reply_Figure 3 in cyan, extracted from (Bourgeois B. et al. 2020, PMID: 32234784, Figure 3D).

We now have included the requested overview in Figure 5D. Only S146 is a canonical SRPK1 phosphosite.

We estimated the final intensity of the phosphorylated peak to approach 100 %. In order to address the point raised by the reviewer, we changed the y-axis label to signal intensity to avoid any potential misinterpretation of the data.

Also, can the authors comment, based on their crystal structure, why phosphorylation of so many residues impairs binding to TNPO3, whereas it favors binding in the case of the canonical motif?

Based on crystal structure comparisons, there is minimal overlap between the canonical phosphorylation sites and the CIRBP^{RSY} phosphorylation sites examined in this study. For example, CIRBP Y160 and A168 are positioned near SRSF1 pS207/pS209, and CIRBP R161 is located close to CPSF6 pS525.

CIRBP Y164 and Y167 are deeply embedded within TNPO3 binding pockets, making their phosphorylation sterically incompatible with binding. Similarly, CIRBP S166 is situated between these pockets, with its side chain oriented toward the TNPO3 surface. This positioning also suggests that phosphorylation at S166 would interfere with binding due to steric hindrance.